# Effects of Melt-Blown Processing Conditions on Nonwoven Polylactic Acid and Polybutylene Succinate

**DOI:** 10.3390/polym15204189

**Published:** 2023-10-23

**Authors:** Patcharee Pratumpong, Thananya Cholprecha, Nanjaporn Roungpaisan, Natee Srisawat, Surachet Toommee, Chiravoot Pechyen, Yardnapar Parcharoen

**Affiliations:** 1Department of Physics, Faculty of Science and Technology, Thammasat University, Khlong Luang, Pathum Thani 12120, Thailand; 2Department of Materials and Textile Technology, Faculty of Science and Technology, Thammasat University, Khlong Luang, Pathum Thani 12120, Thailand; 3Department of Textile Chemistry Engineering, Faculty of Engineering, Rajamangala University of Technology, Khlong Luang, Pathum Thani 12120, Thailandnatee.s@en.rmutt.ac.th (N.S.); 4Industrial Arts Program, Faculty of Industrial Technology, Kamphaeng Phet Rajabhat University, Kamphaeng Phet 62000, Thailand; 5Thammasat University Center of Excellence in Modern Technology and Advanced Manufacturing for Medical Innovation, Thammasat University, Pathum Thani 12120, Thailand; 6Chulabhorn International College of Medicine, Thammasat University, Khlong Luang, Pathum Thani 12120, Thailand

**Keywords:** nonwoven, melt-blown process, polylactic acid (PLA), polybutylene succinate (PBS), air pressure, die-to-collector distance

## Abstract

This research aimed to prepare nonwovens from polylactic acid and polybutylene succinate using the melt-blown process while varying the melt-blown process parameters, including air pressure (0.2 and 0.4 MPa) and die-to-collector distance (15, 30, and 45 cm). Increasing the air pressure and die-to-collector distance resulted in the production of smaller fibers. Simultaneously, the tensile strength was dependent on the polymer, air pressure, and die-to-collector distance used, and the percentage elongation at the break tended to increase with an increasing die-to-collector distance. Regarding thermal properties, the PBS nonwovens exhibited an increased level of crystallinity when the die-to-collector distance was raised, consistent with the degree of crystallinity obtained from X-ray diffraction analysis. Polylactic acid could be successfully processed into nonwovens under all six investigated conditions, whereas nonwoven polybutylene succinate could not be formed at a die-to-collector distance of 15 cm. However, both polymers demonstrated the feasibility of being processed into nonwovens using the melt-blown technique, showing potential for applications in the textile industry.

## 1. Introduction

Nonwoven textiles (or nonwovens) are sheets of web structures joined together by tying fibers or filaments using mechanical, thermal, chemical, or a combination of methods. They are porous, flat sheets that are manufactured solely from individual fibers. Nonwovens can be created without weaving, knitting, and turning the fibers into yarn. Nonwovens have opened a world of innovative possibilities for various industries, such as agriculture, sports, medicine, and furniture. By altering their composition, nonwovens can have specific properties, such as absorbency, liquid repellency, resilience, stretch, softness, strength, flame retardancy, washability, cushioning, filtering, bacterial barriers, and sterility. These properties decide whether the fabrics are suited for any specific application to achieve a good balance between product life cycle and cost [1].

In general, synthetic polymers, such as polyethylene, polypropylene (PP), nylon, and polycarbonate, are used to produce nonwovens. However, these polymers harm ecosystems as they take a long time to decompose and pollute the environment during their production process. To reduce such problems, biodegradable polymers, such as polybutylene succinate (PBS) and polylactic acid (PLA), are used instead. They are produced from natural biodegradable materials without the disposal of toxic substances. In addition, they possess good thermal properties and can be used in various processes, such as film casting, extrusion, blow molding, and fiber spinning.

There are several manufacturing processes for nonwovens, of which the melt-blown (MB) process (MBP) is an interesting one. This is a one-step process in which high-velocity air blows a molten thermoplastic resin from an extruder die tip onto a collector or take-up screen to form a finely fibrous and self-bonding web. A combination of entanglement and cohesive sticking lays the fibers together in the MB web (MBW). The economic advantage of MB technology over alternative systems is due to its ability to form a web straight from a molten polymer without stretching. The vast range of product qualities that MBWs offer also includes random fiber orientation and low to moderate web strength [2,3,4]. Thus, the MBP transforms polymer pellets into a web structure and fiber network in a single step. A schematic view of the MBP is presented in Figure 1.

The MBP uses five constituents: the hopper, extruder, metering pump or gear pump, MB die (MBD), and collector (Figure 2). The polymer, in the form of beads, pellets, chips, or granules, is fed from the hopper into the extruders, where they are heated to melt and reach the desired temperature or viscosity. The melted polymer in the extruder is then transferred by the gear tooth and discharged into the MBD system. The MBD is the most important part because it affects the obtained web structure’s diameter, thickness, tensile strength, and uniformity [5]. The final stage of the MBP is collecting and finishing. When laid down, the fibers are still hot and produce a bonded web ready for the wind-up.

Several factors in the process, such as the die-to-collector distance (DCD) and air pressure, may affect the properties of the obtained fabrics. Therefore, this study aimed to clarify the effect of the DCD and air pressure used in the MBP on the properties of the obtained PLA and PBS nonwovens. The thickness of the MB fiber mat typically decreases as DCD and air pressure rise. Longer collection times are required to attain the same basis weight with a decreasing fiber diameter when air pressure and DCD are increased, resulting in more fibers and thicker layers. The mat’s thickness decreases as the air pressure rises, and smaller pores with a higher packing density are created [6,7]. According to Uppal et al. [8], when air pressure was increased from 70 to 140 kPa, the average diameter of PP fibers generated by an MB pilot line decreased from 590 nm to 520 nm.

Additionally, the DCD plays a significant role in developing the fiber structure and production mechanism. DCD generally produces MB fibers with a bigger fiber diameter and less porosity. The fiber diameter and porosity decrease as the DCD is raised because the attenuation of the fibers is increased. However, a DCD that is excessively large could result in severe flaws, including larger fibers and varied fiber diameters. However, thermoplastic elastomers show phenomenologically different fiber production methods depending on the DCD due to their thermal and elastic relaxation behavior. As a result, the structure and qualities of MB fiber mats may be managed and optimized with DCD [9]. The preparation of nonwovens from PLA and PBS under the MBP was studied in this paper. The effects of the DCD and air temperature on the web structure and its properties were evaluated in terms of the fiber diameter, thickness, thermal properties, mechanical properties, and crystallinity of the PLA and PBS nonwovens for the future development of biodegradable nonwoven fabrics for further applications in the textile industry.

## 2. Materials and Methods

The PLA used in this study was InegoTM biopolymer 3251D resin in pellet form (180 nominal melt flow rate) produced by NatureWorks LLC (Bangkok, Thailand). A standard load of 2.16 kg applied on the resin gave a melt flow index (MFI) of 24 g/10 min at 180 °C. The PBS used was PBS FZ78tm resin in pellet form (200 nominal melt flow rate) produced by PTT MCC BIOCHEM, Thailand. Before use, the PLA and PBS pellets were dried at 80 °C for 4 h via a desiccator to remove excess water.

In this research study, the MBP was performed using a fabric machine (model SR V-N-28, NIHON YUKI brand, SR-RUDER BAMBI) at the textile processing laboratory, Rajamangala University of Technology Thanyaburi. The controlled parameters were a screw rotation speed (TP) of 13.2 rpm and a three-hole round nozzle on the MBD (diameter 0.35 mm). The main production setting parameters are given in Table 1, while the sample codes and conditions of the MBP for PLA and PBS are presented in Table 2.

Morphology of the nonwovens was observed using scanning electron microscopy (SEM; JEOL JSM-6610LV, Oxford X-Max 50, Tokyo, Japan) at magnification 1000× at 10 keV in a low vacuum. Samples were first coated with gold.

Tensile strength and elongation were measured in both machine and cross-direction using an Instron tensile strength tester (ASTM D5034-21 [10]). Ten nonwoven specimens of size 100 × 10 mm were tested at a test speed of 100 mm/min. The specimens were cut randomly from different places and weighed on an electronic balance with an accuracy of 0.005 g and the average of ten readings was calculated. 

Melting behaviors of the MB nonwovens were characterized using differential scanning calorimetry (DSC; NETZSCH DSC 214) on an aluminum pan. All measurements were performed over a 25–200 °C temperature range at a heating rate of 10 °C/min. The temperature and heat flow were calibrated using an indium standard with nitrogen gas purging.

The crystal structure of PLA and PBS nonwovens was characterized using X-ray diffractometry (XRD; Bruker AXS Model D8 Advance), performed in reflection mode with Cu Kα~ radiation at 30 kV and a current of 10 mA over a 2θ range of 5–40°.

## 3. Results and Discussions

### 3.1. Photographs of the PLA and PBS Nonwovens Formed by the MBP

Nonwovens were produced from PLA and PBS by the MBP under six different conditions each (two air pressures for each of three DCDs). The obtained PLA nonwovens, like cotton wool (Figure 3), were fluffy but could not completely form into a fabric sheet. In contrast, the PBS nonwovens prepared at a DCD of 15 cm could not form a fibrous mat but would fuse into a film-like body on the collector [11], as shown in Figure 4.

### 3.2. Thickness and Basis Weight

The thickness and basis weight of the PLA and PBS nonwovens are presented in Figure 5. The thickness of the PLA nonwovens increased when the DCD was increased and slightly decreased when the air pressure was increased. In contrast, as the DCD and air pressure were raised, the thickness and basis weight of the PBS nonwovens decreased. The drag forces near the collector during fiber laydown can be used to illustrate how DCD and air pressure affect the process. At two key MBP positions close to the MBD, drag forces are significant. The air speed is very fast due to the high air pressure, while the fiber speed is prolonged, so the aerodynamic drag is large. Near the collector, the air speed is still quite fast. Nevertheless, during laydown on the collector, the fiber speed falls to zero, considerably increasing aerodynamic drag. The air speed at the collector and the drag force acting on the fibers during laydown are reduced since the collector is situated farther from the air source at a larger DCD. In contrast, the thickness and basis weight of PLA nonwovens tended to increase due to the different polymeric materials. The main factors affecting the fibers’ characteristics are the velocity, polymer type, and MFI, even if they are produced under the same conditions.

### 3.3. Microstructure of the PLA and PBS Nonwovens Formed by the MBP

Representative SEM images of the PLA nonwovens found at different air pressures and DCDs are presented in Figure 6. Similar long fibers were found under all six MBP conditions, but the fibers’ diameters differed according to the forming conditions.

The diameter of each sample type is as follows: (i) DCD 15 cm, air pressure of 0.2 MPa: the fibers are arranged in an orderly manner and approximately 3.19 µm in diameter; (ii) DCD 15 cm, air pressure of 0.4 MPa: long fibers are arranged in an orderly manner but approximately 2.95 µm in diameter; (iii) DCD 30 cm, air pressure 0.2 MPa: fiber diameter of about 2.79 µm; (iv) DCD 30 cm, air pressure 0.4 MPa: fiber diameter about 3.46 µm, but tiny particles were found on the fibers, possibly due to excessive air pressure causing the fibers to break, create droplets, and attach to the fibers; (v) DCD 45 cm, air pressure 0.2 MPa: long fibers are bent, and the sample is arranged as disorganized fibers of approximately 2.77 µm diameter; (vi) DCD 45 cm, air pressure 0.4 MPa: fibers are approximately 2.25 µm in diameter.

Thus, increasing the DCD at an air pressure of 0.2 MPa decreased the fiber diameter, whereas, at 0.4 MPa, it first increased the diameter to the largest seen (3.46 µm) when increasing the DCD from 15 to 30 cm but then decreased it upon further rising to 45 cm to the smallest diameter (2.25 µm). Likewise, increasing the air pressure from 0.2 to 0.4 MPa decreased the fiber diameter at a DCD of 15 or 45 cm but increased it at a DCD of 30 cm.

For PBS, nonwoven PBS sheets could not be formed from the MBP at a DCD of 15 cm with an air pressure of either 0.2 or 0.4 MPa. Rather, they developed a large lump of PBS that merged into a homogeneous state (Figure 7) due to the distance being too close for a relatively viscous polymer PBS. Due to its high viscosity, PBS takes longer to crystallize than PLA. The melted PBS slowly transforms into tiny filaments or fibers after exiting the MBD. Still, at a DCD of 15 cm, the polymer’s crystallization time is too short, and on the collector, the fibers are still very hot with a viscous character and merge into a homogeneous state.

Figure 8 shows SEM images of the PBS nonwoven formed by the MBP under the other four conditions (DCD of 30 and 45 cm each with an air pressure of 0.2 and 0.4 MPa), where the fibers had a similar length and were relatively large in diameter, especially at a DCD of 30 cm and air pressure of 0.2 MPa. However, the fiber diameter was reduced when the air pressure or the DCD increased. 

A summary of the effects of the DCD and air pressure on the formation of PBS and PLA sheets by the MBP is as follows. Increasing the DCD decreased the average fiber diameter, and increasing the air pressure reduced the degree of fiber entanglement. Still, the average fiber diameter of PBS nonwovens decreased. The average fiber diameter of PLA nonwovens, in contrast, is unclear because the MBP nonwovens showed at least three types of interfiber bonding or fiber entanglements, including thermal sticking, branching, interlacing, and the degree of fiber entanglement, which are used to describe an MBW. The entangled fibers could be fused and lose their individuality if they are not sufficiently quenched, as seen in Figure 8.

### 3.4. Mechanical Properties

The mechanical properties of the tensile strength (tensile strength measures a material’s resistance to stretching or pulling forces. It is expressed in megapascals (MPa), and a higher value indicates greater strength) and % elongation at break (% elongation at break measures a material’s ability to stretch before it breaks. It indicates the flexibility of the material. Higher values suggest greater flexibility) of the PBS and PLA nonwoven samples formed by the MBP under different air pressure and DCD values are summarized in Table 3. The PBS nonwovens (formed from DCDs of 30 and 45 cm only, as none could be formed from a DCD of 15 cm) showed a higher tensile strength than the PLA nonwovens owing to their larger diameter fibers.

For the PLA nonwovens, the highest tensile strength (0.75 MPa) was found in that formed with a DCD of 15 cm and air pressure of 0.2 MPa. Increasing the DCD at the same air pressure reduced the tensile strength markedly to 0.15 and 0.11 MPa at 30 and 45 cm, respectively, the latter having the lowest tensile strength of all six PLA sample types. Increasing the air pressure from 0.2 MPa to 0.4 MPa reduced the tensile strength from 0.75 to 0.68 MPa at a DCD of 15 cm but increased it from 0.15 to 0.32 MPa and from 0.11 to 0.13 MPa at DCDs of 30 and 45 cm, respectively.

The PBS nonwovens (DCD of 30 and 45 cm only) that formed at a DCD of 30 cm and air pressure of 0.2 MPa had a tensile strength of 0.75 MPa. Increasing the DCD at this air pressure to 45 cm reduced the tensile strength to 0.39 MPa. In contrast, increasing the air pressure increased the tensile strength from 0.75 to 0.95 MPa (the highest tensile strength) at a DCD of 30 cm and from 0.39 to 0.57 MPa at a DCD of 45 cm. 

Overall, the tensile strength of the PLA nonwovens was strongly influenced by the DCD, being highest at a DCD of 15 cm and markedly reduced at higher DCD values, whereas increasing the air pressure had only a slight increase in the tensile strength. Thus, increasing the DCD may negatively affect the fiber-to-fiber bond strength. However, the air pressure was less consistent, decreasing the tensile strength at a DCD of 15 cm, and strongly and barely increasing it at DCDs of 30 and 45 cm, respectively. This is mainly because when the air pressure is too high, the fibers become very disheveled and entangled before they land on the collector, leading to an uneven web that affects the tensile strength. In contrast, increasing the air pressure strongly increased the tensile strength of the PBS nonwovens.

Concerning the elongation at break of the PLA and PBS nonwovens formed by the MPB (Table 3), the PLA nonwovens showed a higher % elongation at break than the PBS nonwovens owing to the properties of the two polymers. For the % elongation at break of the PLA nonwovens, increasing the air pressure from 0.2 MPa to 0.4 MPa decreased the % elongation at break from 13.22 to 2.63% at a DCD of 15 cm and from 23.32 to 12.45% at a DCD of 30 cm. However, at a DCD of 45 cm, the air pressure increase from 0.2 to 0.4 MPa slightly increased the % elongation at break from 27.24% to 30.58%, which was the highest value for the six different types of PLA nonwoven samples.

For the PBS nonwovens (DCDs of 30 and 45 cm only), increasing the DCD from 30 to 45 cm only slightly increased the % elongation at break from 6.86 to 7.16% at 0.2 MPa air pressure, but the increase was more pronounced at 0.4 MPa air pressure (6.85 to 10.63%). Increasing the air pressure from 0.2 to 0.4 MPa did not affect the % elongation at break at a DCD of 30 cm (remained at the lowest value of 6.85–6.86%) but increased the % elongation at break at a DCD of 45 cm from 6.85 to 10.63%. Thus, increasing the DCD may positively affect the flexibility of nonwovens. The applied air pressure had an uneven effect on the % elongation at break, where increasing the air pressure increased the % elongation at break of the PBS nonwovens but with no impact or only a slight decrease in the % elongation at break of the PLA nonwovens.

### 3.5. Thermal Properties

The thermal behavior of the pure PLA and PBS pellets and the PLA and PBS nonwovens were measured using DSC analysis, with representative curves shown in Figure 9 and summarized in Table 4. The glass transition temperature (T_g_) of pure PLA pellets was around 68.10 °C. In contrast, it was decreased to about 60.10 °C in the PLA nonwovens, remaining almost unchanged concerning the different air pressure and DCDs used in their formation by the MBP. Thus, the MBP affected the motion of the PLA molecule chains, but this was independent of the exact DCD and air pressure used. The DSC curve of all six different MBP samples of PLA showed an exothermic peak corresponding to the cold crystallization temperature (T_cc_) of the process, which is due to the mobility and rearrangement of PLA molecule chains. Still, the T_cc_ varied between the different PLA nonwoven samples. The pure PLA pellets and PLA nonwovens exhibited multiple crystallization behaviors, but the T_cc_ peak did not appear in the pure PLA pellets because crystallization occurs in the PLA that has undergone the MBP (PLA nonwovens) and the different MBP conditions lead to different T_cc_ values. The crystalline melting temperature (T_m_) was approximately the same in all six types of PLA nonwovens (about 167 °C), slightly different from the T_m_ of the PLA pellets. The melt recrystallization of crystallites with differing thermal stabilities has been connected to explaining the T_m_ peak. The endothermal peaks are attributed to three different types of crystals, the re-melting of crystallites formed during recrystallization, and the annealing peak, which marks the change of the rigid amorphous fraction (RAF) from a solid-like RAF to a liquid-like RAF. These findings are in line with those of earlier research [12].

Thermal analysis using DSC revealed that the pure PBS and pellets and the various PBS and PLA nonwovens usually displayed distinct T_cc_ and T_m_ transitions for the hard and soft segments, respectively, as plotted in Figure 9 and analyzed to reveal the effect of the DCD and air pressure on the thermal behaviors of the obtained PBS and PLA nonwovens. They exhibited a similar thermal behavior for the PBS pellets and derived nonwovens, where the T_cc_ of pure PBS pellets was 95.4 °C and increased to around 97–98 °C for the PBS nonwovens. Naturally, the polymer chains with a lower molecular weight should crystallize first because their mobility and diffusion rate are higher at any given temperature. Thus, the MBP affected the motion of the PBS molecule chains, with changes in the DCD and air pressure having a slight and no significant effect on the T_cc_ of PBS nonwovens [13]. Subsequent melting curves of the pure PBS pellets and the different PBS nonwovens revealed that the T_m_ of the pure PBS pellets (116.4 °C) was decreased in the PBS nonwovens to around 114 °C. That is, the MBP affected the melting of PBS. The α-crystal’s initial formation is associated with the first melting peak, and the recrystallized crystal is related to the second melting peak. The α- and β-crystal combination was connected to melting peaks. The intricate appearance of two melting peaks is most likely caused by the interaction of the β- to α-phase transition and the recrystallization mechanism. It is also believed that the phase transition mechanism that took place during the heating process caused the T_m_ of the β-crystal to be considerably higher than that of the α-one [14].

The heating DSC thermograms of the studied materials are summarized in Table 3 in terms of the estimated variation in the T_g_, T_m_, crystallization temperature (T_c_), T_cc_, and degree of crystallinity (X_c_) for each studied PLA and PBS nonwoven sample. The X_c_ was calculated using the following equation:(1)$$\%X_{c}=\frac{∆H_{m}-{∆H}_{cc}}{{∆H{}^{\circ}}_{f}}\times 100,$$
where ΔH_m_ is the melting enthalpy, ΔH_cc_ is the cold crystallization enthalpy, and ΔH°_f_ is the melting enthalpy of the 100% crystalline polymer (equal to 93.1 J/g for PLA [15] and 110.3 J/g for PBS [16]).

### 3.6. Physical Property (XRD Analysis)

Representative XRD patterns of the PLA and PBS nonwovens formed by the MBP under different conditions are shown in Figure 10. The DSC study was in agreement with the PLA nonwovens’ diffraction patterns, which showed diffuse rather than distinct crystal diffraction peaks, indicating partial crystallization or an amorphous condition. The high-intensity diffraction peaks were identified at 16.2° and the second intensity at 31.5°. These diffraction peaks are consistent with previous work [17]. The XRD patterns of the PBS nonwovens were more crystalline than those of the PLA nonwovens. The curves of PBS-1 and PBS-2 (DCD of 15 cm and air pressure of 0.2 and 0.4 MPa, respectively) showed a diffraction peak intensity at a 2θ of 22.8°, and lower intensity peaks at 2θ of 19.8° and 29.1°. They were significantly less crystalline than the PBS nonwovens formed under other MBP conditions because the forming distance (DCD) of 15 cm was too close. So, the PBS polymers could not crystallize in time during the formation of the fibers and the sheet. This resulted in them being unable to form a complete nonwoven sheet and less crystalline than those with longer DCD values. Increasing the DCD to 30 and 45 cm (PBS-3–PBS-6) resulted in a higher crystallinity (Figure 10b). The DCD and air pressure had an insignificant impact on the crystallinity of the PBS nonwovens. The XRD curves of PBS-4 and PBS-6 (DCDs of 30 and 45 cm, respectively, at an air pressure of 0.4 MPa) had a markedly higher intensity peak than at 0.2 MPa air pressure (PBS-3 and PBS5). The diffraction peak intensity at a 2θ of 22.5° and the lower intensity diffraction peaks at 2θ of 19.8°, 21.6°, and 29.1° correspond to the diffraction peaks of PBS fibers in a previous report [18]. Thus, increasing the DCD and the air pressure increased the crystalline content of PBS nonwovens.

## 4. Conclusions

The MBP consists of extruding the polymer through the small orifices of a die, driven with the aid of a high-velocity hot air jet to generate continuous, fine polymeric fibers in the form of a nonwoven sheet.

In this study, the influence of air pressure (0.2 and 0.4 MPa), DCD (15, 30, and 45 cm), and polymer (PLA and PBS) on the thermal properties, morphology, mechanical properties, and crystal structure of the obtained nonwoven fabric formed by the MBP was evaluated. The results can be summarized as follows:The PLA and PBS nonwovens prepared via the MBP comprised small fibers (µm diameter scale). PLA nonwovens had a smaller fiber size than PBS nonwovens under all paired MBP conditions examined;For the polymer type, PBS required a higher melting temperature throughout the process than PLA. Due to the two polymers’ different rheology indices and polymer behavior, the PBS nonwovens had a higher crystallinity and tensile strength than the PLA nonwovens. However, the % elongation at break of the PLA nonwovens was better than the PBS nonwovens due to their smaller fiber size;Using an air pressure of 0.4 MPa in the MBP gave both PLA and PBS nonwovens with smaller fiber sizes, and the crystalline content was greater at a lower air pressure (0.2 MPa);The DCD influenced the nonwoven PBS and PLA fibers. Although PBS could not be formed into a complete fiber sheet or nonwoven at a DCD of 15 cm, when the DCD was increased to 30 and 45 cm, the PBS could be completely formed into a fabric sheet with an increasing crystal volume and decreased fiber size as the DCD grew. For PLA nonwovens, the fiber size is also reduced with increasing DCD.

Therefore, the influential factors in the MBP, in terms of the properties of the obtained nonwoven fabrics, are the type of polymer, air pressure, and DCD. All influenced the fiber size of the nonwovens, thermal properties, mechanical properties, and crystal structure. Nevertheless, the MBP can be employed to prepare nonwovens from both PLA and PBS and be developed for further use in the textile industry.

## Figures and Tables

**Figure 1 polymers-15-04189-f001:**
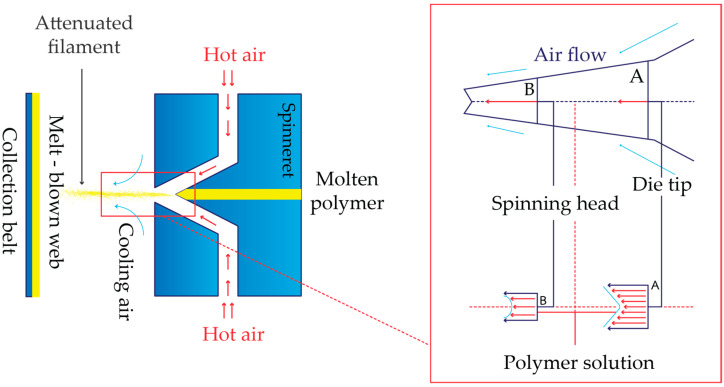
Schematic diagram of the MBP principles.

**Figure 2 polymers-15-04189-f002:**
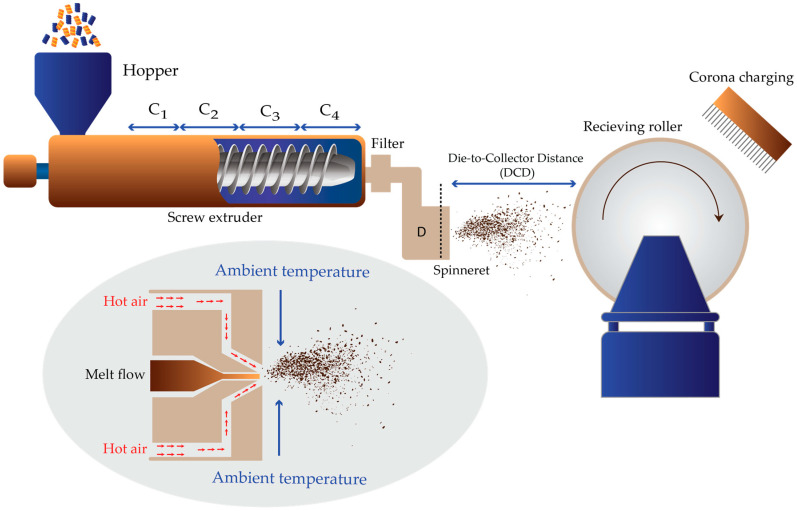
Schematic diagram showing the MBP equipment with the extruder and its zones: zone C1—the feed zone; zones C2 and C3—the transition zone; zone C4—the metering; and zone D—the MB.

**Figure 3 polymers-15-04189-f003:**
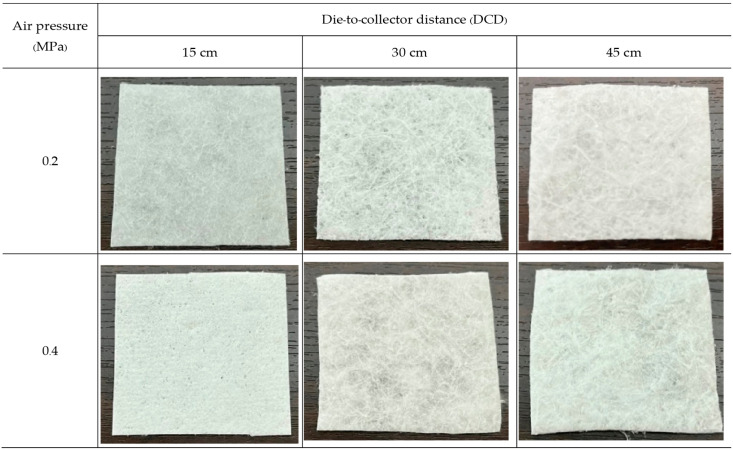
Photographs of the PLA nonwovens formed by the MBP at various air pressure and DCD values.

**Figure 4 polymers-15-04189-f004:**
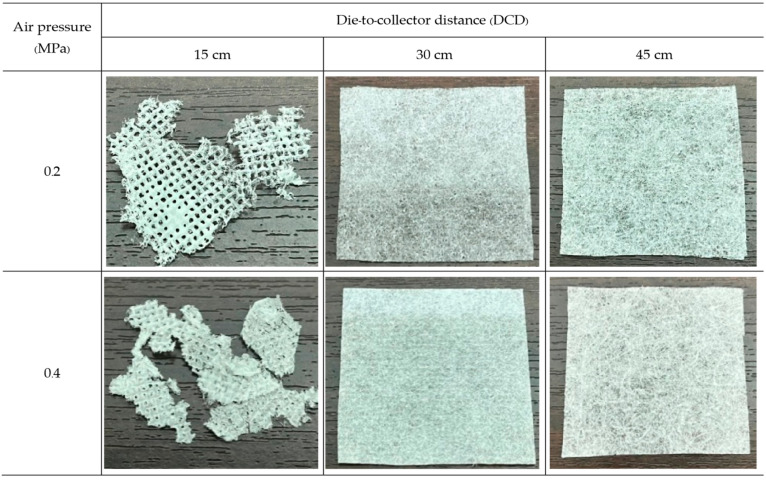
Photographs of the PBS nonwovens formed by the MBP at various air pressure and DCD values.

**Figure 5 polymers-15-04189-f005:**
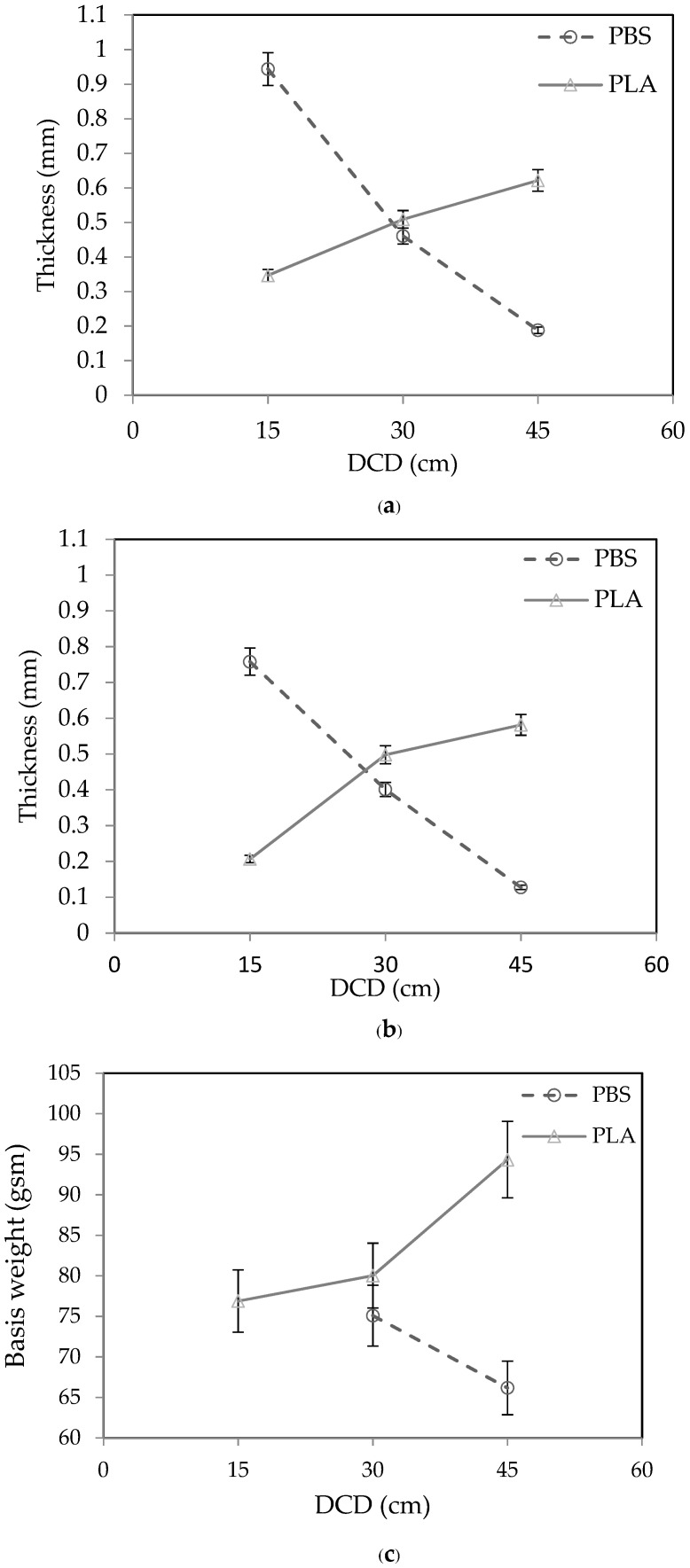
Influence of the DCD on the (**a**,**b**) fabric thickness and (**c**,**d**) basis weight of the PLA and PBS nonwovens formed by the MBP with an air pressure of (**a**,**c**) 0.2 MPa and (**b**,**d**) 0.4 MPa. Data are shown as the mean ± 1 standard deviation, derived from 3 independent samples. Note that PBS nonwovens could not be formed at a DCD of 15 cm.

**Figure 6 polymers-15-04189-f006:**
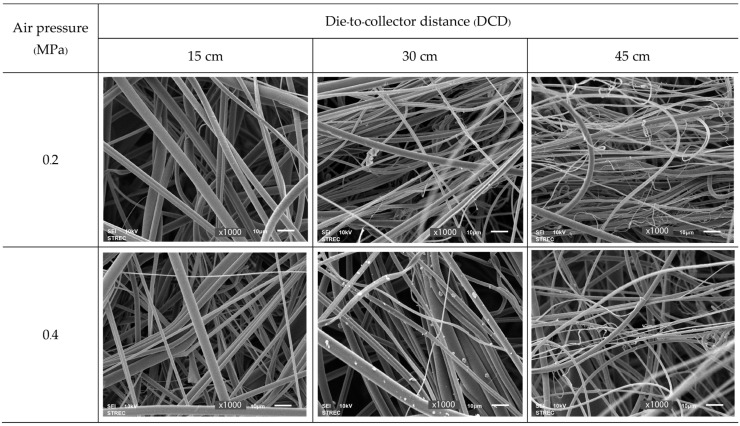
Representative SEM images of the PLA nonwovens formed by the MBP at various air pressure and DCD values. (Magnification ×1000).

**Figure 7 polymers-15-04189-f007:**
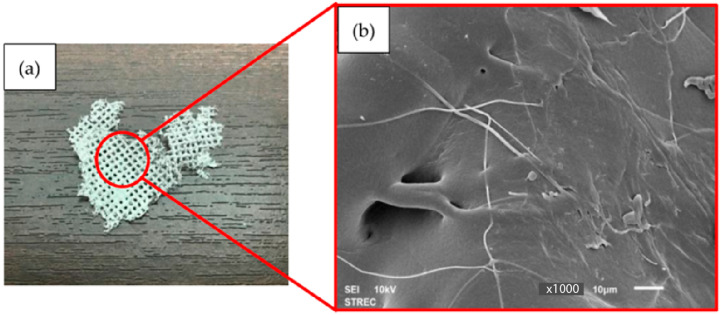
Representative (**a**) photographic and (**b**) SEM images (magnification: ×1000) of a PBS nonwoven sample formed by MBP at a DCD of 15 cm and air pressure of 0.2 MPa.

**Figure 8 polymers-15-04189-f008:**
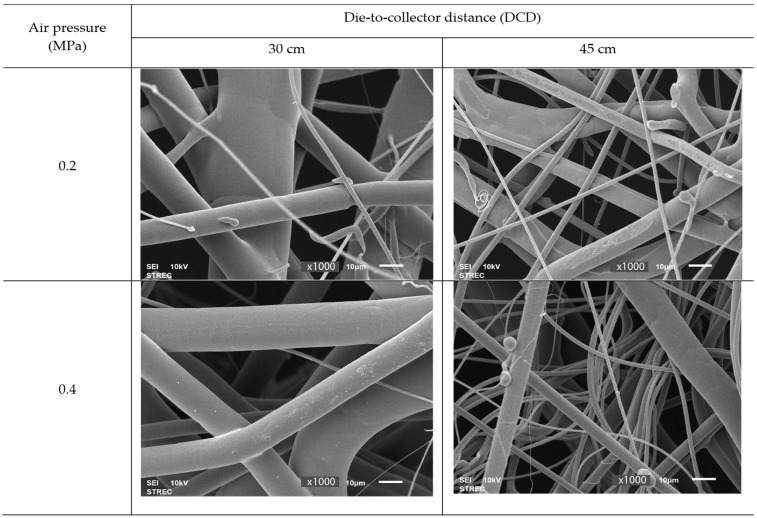
Representative SEM images (magnification: ×1000) of PBS nonwovens formed by the MBP at various air pressure and DCD values.

**Figure 9 polymers-15-04189-f009:**
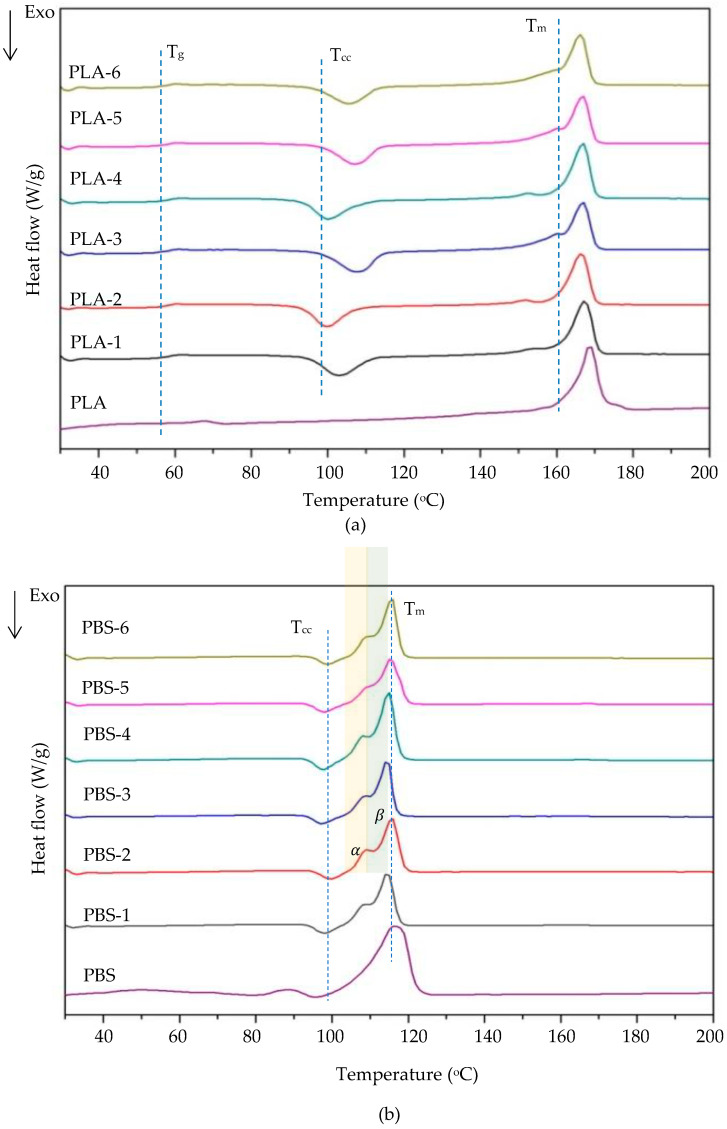
Representative DSC curves of the (**a**) PLA and (**b**) PBS nonwovens formed via the MBP at various air pressure and DCD values. Sample codes are given in Table 2.

**Figure 10 polymers-15-04189-f010:**
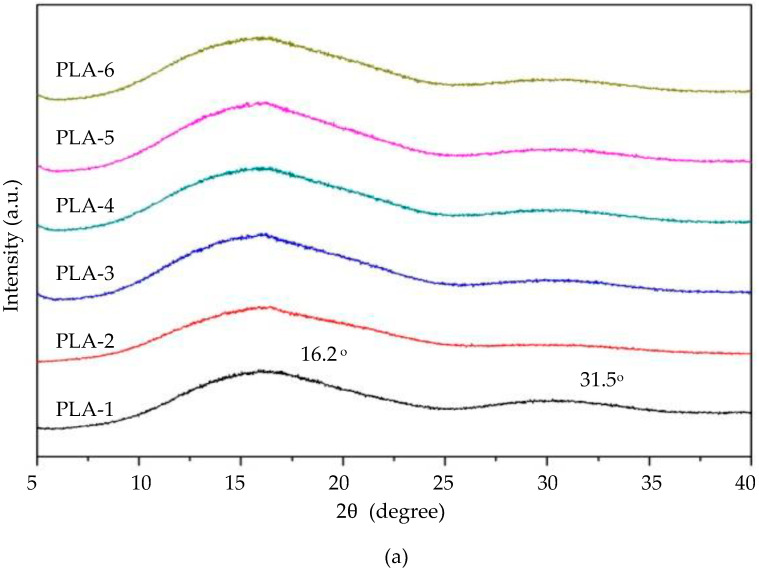
Representative XRD patterns of the (**a**) PLA and (**b**) PBS nonwovens formed by the MBP under different DCD and air pressures.

**Table 1 polymers-15-04189-t001:** Main production setting parameters.

Parameter		Polymers	
		Polylactic Acid (PLA)	Polybutylene Succinate (PBS)
Extruder zone	C_1_ (°C)	170	170
	C_2_ (°C)	230	230
	C_3_ (°C)	240	250
	C_4_ (°C)	240	270
Die temperature zone D (°C)		250	275
Air temperature (°C)		260	260
Collector diameter (cm)		45	45
Collector speed (m/min)		2.5	2.5

**Table 2 polymers-15-04189-t002:** Sample code and production factors.

Sample Code		Conditions	
		Air Pressure (MPa)	DCD (cm)
Polylactic acid (PLA)	PLA-1	0.2	15
	PLA-2	0.4	15
	PLA-3	0.2	30
	PLA-4	0.4	30
	PLA-5	0.2	45
	PLA-6	0.4	45
Polybutylene succinate (PBS)	PBS-1	0.2	15
	PBS-2	0.4	15
	PBS-3	0.2	30
	PBS-4	0.4	30
	PBS-5	0.2	45
	PBS-6	0.4	45

**Table 3 polymers-15-04189-t003:** Mechanical properties regarding the tensile strength and % elongation at break of the PLA and PBS nonwovens formed by the MBP at various DCD and air pressure values. Data are the mean ± 1 SD, derived from 3 independent repeats.

Condition			Tensile Strength (MPa)	% Elongation
Polymers	Air Pressure (MPa)	DCD (cm)		
Polylactic acid (PLA)	0.2	15	0.75 ± 0.04	13.22 ± 0.66
	0.4	15	0.68 ± 0.03	2.63 ± 0.13
	0.2	30	0.15 ± 0.01	23.32 ± 1.16
	0.4	30	0.32 ± 0.02	12.45 ± 0.62
	0.2	45	0.11 ± 0.01	27.24 ± 1.36
	0.4	45	0.13 ± 0.01	30.58 ± 1.53
Polybutylene succinate (PBS)	0.2	15	-	-
	0.4	15	-	-
	0.2	30	0.25 ± 0.01	6.86 ± 0.34
	0.4	30	0.95 ± 0.05	6.85 ± 0.34
	0.2	45	0.39 ± 0.02	7.16 ± 0.36
	0.4	45	0.57 ± 0.03	10.63 ± 0.53

Note: PBS nonwovens could not be formed at a DCD of 15 cm.

**Table 4 polymers-15-04189-t004:** Thermal properties of the PLA and PBS nonwovens formed from the MBP under different conditions.

Conditions		Polylactic Acid (PLA)					Polybutylene Succinate (PBS)			
Air Pressure (MPa)	DCD (cm)	T_g_ (°C)	T_m_ (°C)	T_cc_ (°C)	$∆H_{m}$(J/g)	$X_{c}$(%)	T_cc_ (°C)	T_m_ (°C)	$∆H_{m}$(J/g)	$X_{c}$(%)
Neat		68.1	169.2	-	41.3	-	95.4	116.4	77.9	61.93
0.2	15	60.1	167.1	105.8	54.7	15.80	98.1	114.5	72.0	56.80
0.4	15	60.2	167.3	100.7	51.3	21.03	98.7	114.6	80.4	64.53
0.2	30	60.0	167.3	109.0	42.4	10.05	97.3	114.5	70.9	55.51
0.4	30	60.1	167.0	100.9	44.3	13.34	97.4	114.8	75.7	59.61
0.2	45	60.2	167.0	108.9	50.3	18.93	97.1	114.4	72.3	56.34
0.4	45	60.0	167.2	106.5	52.23	20.91	97.5	114.6	80.2	64.88

## Data Availability

The data presented in this study are available on request from the corresponding author.

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
