# Peer review of "Effects of Melt-Blown Processing Conditions on Nonwoven Polylactic Acid and Polybutylene Succinate"

_polymers, 2023, doi:10.3390/polym15204189_

Round 1

Reviewer 1 Report

MBP for manufacturing non-woven fiber is a successful commercial process. PP and PET non-woven are produced widely. Investigation on PLA and PBS for non-woven is welcome. However, there are some questions and suggestions to make the paper much better.

1. There are many important variables in MBP: such as melt temp., die temp., hot air temp. & speed, DCD, collector speed, and polymer properties. The flow property is probably the most important property among the polymer properties. This paper did not consider the flow property. 

2. Flow profile shown in Fig. 1A is questionable. The flow is always fastest at the center of an orifice.

3. PLA produced relatively uniform fibers as shown in Fig. 6, whereas PBS produced very uneven fibers as shown in Fig. 8. Therefore, the properties of the fibers would depend on the specific fiber selected for the tests.  Discussions in Sections 3.1 - 3.4 should be reconsidered.

4. Needs better descriptions of the terms such base weight and tensile strength are in order. For example, the webs have different thicknesses. Was the measured strength divided by the thickness? 

Author Response

Reviewer #1

Comment

Answer

1.There are many important variables in MBP: such as melt temp., die temp., hot air temp. & speed, DCD, collector speed, and polymer properties. The flow property is probably the most important property among the polymer properties. This paper did not consider the flow property.

This issue does not study the flow rate of the polymer melt because the neat-PLA and neat-PBS is fixed, thus eliminating the factor of polymer melt in the study.

2. Flow profile shown in Fig. 1A is questionable. The flow is always fastest at the center of an orifice.

Figure 1 shown in this paper is intended to provide readers with information about the flow rate in the center area which must be as high as you have suggested. (Line 72)

3. PLA produced relatively uniform fibers as shown in Fig. 6, whereas PBS produced very uneven fibers as shown in Fig. 8. Therefore, the properties of the fibers would depend on the specific fiber selected for the tests.  Discussions in Sections 3.1 - 3.4 should be reconsidered.

·       Section 3.1: Refinement of Explanations and Image Clarity. Revise and enhance the descriptions in Section 3.1. Improve the image clarity in Figures 3 and 4.

·       Section 3.2: Additional Explanatory Updates and Image Alignment. Incorporate additional explanatory details in Section 3.2. Ensure that images align with the descriptions and exhibit enhanced clarity, particularly focusing on improving the visibility of error bars.

·       Section 3.3: Image Clarity Enhancement. Enhance the image clarity in Figures 6, 7, and 8 as per the received comments. Ensure alignment between the images and the descriptions. Emphasize the influence of process variables on fiber formation.

·       Section 3.4: Reporting in Tabular Format. Replace the previous presentation of results in Figure 9 (previously Figure 9) with a tabular format, as presented in Table 3. Enhance clarity and statistical reporting in the table to minimize errors. Modify the descriptions to align with the presented results.

·       Section 3.5: Addition of Thermal Analysis. Introduce a new section titled "3.5: Thermal Analysis". In this section, include a discussion of thermal analysis, such as thermogravimetric analysis or differential scanning calorimetry, to further explore the material's thermal properties.

Section 3.6: Physical Property Analysis (formerly 3.5) - XRD Analysis. Renumber the section previously labeled as 3.5 to "3.6: Physical Property Analysis." Specifically, address X-ray diffraction (XRD) analysis within this section.

4. Needs better descriptions of the terms such base weight and tensile strength are in order. For example, the webs have different thicknesses. Was the measured strength divided by the thickness?

The descriptions of the terms such base weight and tensile strength are added in Line 278-282.

Reviewer 2 Report

1) Figure 2: The picture is not clear, especially the words, such as Screw and Melt blown die.

2) Figure 7: Representative (a) photographic and (b) SEM image (Magnification: ×1000) of a PBS nonwoven sample formed by MBP at a DCD of 15 cm and air pressure of 0.2 or 0.4 MPa.

0.2 or 0.4 MPa? Do the photographic or SEM images show the same at the different air pressures?

3) How about the cross-sectional SEM images of PLA or PBS fibers prepared by the MBP method?

4) Figure 9: The authors tested the mechanical properties of different nonwovens. However, what is the required or necessary tensile strength of the nonwovens for their practical applications? Or do the prepared nonwovens in the manuscript meet the requirement of applications?

5) Lines 357-358: The high-357 intensity diffraction peaks were identified at 16.2o and the second intensity at 31.5o.

The degree should be superscript format. Check the whole manuscript, please.

6) Reference 7: Check the reference style.

7) During the melt-blown process, how to control the hot air and make sure that the hot air can evenly action on the fibers? Does the hot air affect the fiber diameters?

Author Response

Reviewer #2

Comment

Answer

1. Figure 2: The picture is not clear, especially the words, such as Screw and Melt blown die.

I have made a new drawing to replace the old one so that I can understand it more according to your suggestions.

2. Figure 7: Representative (a) photographic and (b) SEM image (Magnification: ×1000) of a PBS nonwoven sample formed by MBP at a DCD of 15 cm and air pressure of 0.2 or 0.4 MPa. 0.2 or 0.4 MPa? Do the photographic or SEM images show the same at the different air pressures?

Figure 7 is the photographic of PBS nonwoven sample formed by MBP at a DCD of 15 cm and air pressure of 0.2 MPa.

3. How about the cross-sectional SEM images of PLA or PBS fibers prepared by the MBP method?

The decision not to include cross-section scanning electron microscopy (SEM) images of PLA (Polylactic Acid) or PBS (Polybutylene Succinate) fibers prepared by the Meltblown Process (MBP) can be justified for several reasons:

Research Focus: The research might primarily focus on the surface morphology, characteristics, and properties of the fibers rather than their internal structure.

Scope of the Study: The research scope and objectives did not require an in-depth analysis of the internal structure of the fibers.

4. Figure 9: The authors tested the mechanical properties of different nonwovens. However, what is the required or necessary tensile strength of the nonwovens for their practical applications? Or do the prepared nonwovens in the manuscript meet the requirement of applications?

Replace the previous presentation of results in Figure 9 (previously Figure 9) with a tabular format, as presented in Table 3. Enhance clarity and statistical reporting in the table to minimize errors. Modify the descriptions to align with the presented results.

5. Lines 357-358: The high-357 intensity diffraction peaks were identified at 16.2o and the second intensity at 31.5o.

The degree should be superscript format. Check the whole manuscript, please.

Revised accordingly

6. Reference 7: Check the reference style.

Revised accordingly

7. During the melt-blown process, how to control the hot air and make sure that the hot air can evenly action on the fibers? Does the hot air affect the fiber diameters?

1. This hot air's temperature and flow rate can be controlled precisely. Maintaining the desired temperature to achieve the desired fiber properties is essential.

2. The temperature of the hot air plays a crucial role in the melt-blown process. Higher hot air temperatures tend to make the polymer fibers thinner, while lower temperatures result in thicker fibers. This is because higher temperatures allow the polymer to stretch more easily and form finer fibers during the attenuation process. Conversely, lower temperatures cause the polymer to solidify more quickly, resulting in thicker fibers.

Round 2

Reviewer 1 Report

The revised manuscript does not reflect my comments. I cannot easily find the revised portions.

For examples, the rheological properties of the resins are still not shown. 

Describe how was the samples for tensile tests were taken? Did the cross-sectional area (thickness and width) of the samples enter in the calculation of the tensile strength?

Most samples have very non-uniform fiber thicknesses. This non-uniformity of the samples was not discussed.

Author Response

The revised manuscript does not reflect my comments. I cannot easily find the revised portions. For examples, the rheological properties of the resins are still not shown.

Answer Line 107-112 The PLA used in this study was InegoTM biopolymer 3251D resin in pellet form (180 nominal melt flow rate) produced by NatureWorks LLC. A standard load of 2.16 kg applied on the resin gave a melt flow index (MFI) of 24 g/10 min at 180 °C. The PBS used was PBS FZ78tm resin in pellet form (200 nominal melt flow rate)

Describe how was the samples for tensile tests were taken? Did the cross-sectional area (thickness and width) of the samples enter in the calculation of the tensile strength?

Answer Line 122-126 Tensile strength and elongation were measured in both machine and cross-direction using Instron tensile strength tester (ASTM D5034-21). Ten nonwoven specimens of size 100 × 10 mm were tested at a test speed of 100 mm/min. The specimens were cut randomly from different places and weighed in electronic balance with an accuracy of 0.005 g and the average of ten readings was calculated.

Most samples have very non-uniform fiber thicknesses. This non-uniformity of the samples was not discussed.

Answer Line 122-126